# C3PO: Learning to Achieve Arbitrary Goals via Massively Entropic Pretraining

## Abstract

Given a particular embodiment, we propose a novel method (C3PO) that learns policies able to achieve any arbitrary position and pose. Such a policy would allow for easier control, and would be re-useable as a key building block for downstream tasks. The method is two-fold: First, we introduce a novel exploration algorithm that optimizes for uniform coverage, is able to discover a set of achievable states, and investigates its abilities in attaining both high coverage, and hard-to-discover states; Second, we leverage this set of achievable states as training data for a universal goal-achievement policy, a goal-based SAC variant. We demonstrate the trained policy's performance in achieving a large number of novel states. Finally, we showcase the influence of massive unsupervised training of a goal-achievement policy with state-of-the-art pose-based control of the Hopper, Walker, Halfcheetah, Humanoid and Ant embodiments.

## 1 Introduction

Reinforcement learning (RL) has shown great results in optimizing for single reward functions (Mnih et al., 2013; Silver et al., 2017; Vinyals et al., 2019), that is when a controller has to solve a specific task and/or the task is known beforehand. If the task is not known *a priori*, or is likely to be often re-configured, then re-training a new policy from scratch can be very expensive and looks as a waste of resources. In the case of multipurpose systems deployed in contexts where they will likely be required to perform a large range of tasks, investing significant resources into training a high-performance general goal-based controller beforehand makes sense. We propose an approach allowing for training a *universal goal achievement policy*, a policy able to attain any arbitrary state the system can take. Goal-conditioned RL is such a setting where a single policy function can be prompted to aim for a particular goal-state (Kaelbling, 1993; Schaul et al., 2015). One important issue with goal-conditioned RL is that goals that are useful for training the policy are generally unknown, and even in the case of humans this is a key part of learning general controllers (Schulz, 2012; Smith and Gasser, 2005). Several approaches exist in the literature. Adversarial methods build out a goal-based curriculum (Mendonca et al., 2021; Eysenbach et al., 2018; OpenAI et al., 2021; Florensa et al., 2018) through various ad-hoc 2-player games. Other recent approaches (Kamienny et al., 2021; Campos et al., 2020) explicitly optimize for uniform state coverage with the goal of learning a general goal-conditioned policy, but are still tied to learning a policy function to actually implement the exploration strategy in the environment. Although not explicitly geared towards goal-based learning, many reward-free RL (Laskin et al., 2021) approaches are geared towards learning policies that provide good state coverage (Bellemare et al., 2016; Ostrovski et al., 2017; Burda et al., 2018; Houthooft et al., 2016; Badia et al., 2020), however primarily with the intent of fine-tuning the learnt exploration policy rather than leveraging its state coverage. Our proposed approach, *Entropy-Based Conditioned Continuous Control Policy Optimization* (C3PO), is based on the hypothesis that disentangling the exploration phase from the policy learning phase can lead to simpler and more robust algorithms. It is composed of two steps:

- **Goal Discovery:** generating a set of achievable states, as diverse as possible to maximize coverage, while being as uniform as possible to facilitate interpolation.

- **Goal-Conditioned Training:** leveraging these states to learn to reach arbitrary goals.

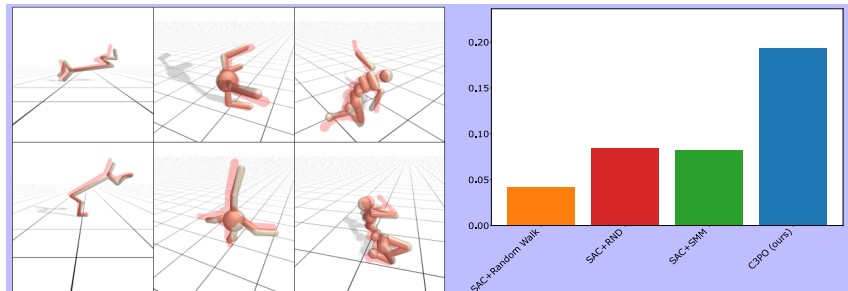

Figure 1: Left: Examples of the C3PO (in beige) achieving a pose (in red). Right: Entropy-Weighted Goal Achievement of C3PO vs. other methods, averaged over Walker, Hopper, Halfcheetah and Ant (see Section 3.3).

To address the goal discovery step in C3PO, we propose the Chronological Greedy Entropy Maximization (ChronoGEM) algorithm, designed to exhaustively explore reachable states, even in complex high dimensional environments. ChronoGEM does not rely on any form of trained policy and thus doesn't require any interaction with the environment to learn to explore. Instead it uses a highly-parallelized random-branching policy to cover the environment, whose branching tree is iteratively re-pruned to maintain uniform leaf coverage. This iterative pruning process leverages learnt density models and inverse sampling to maintain a set of leaf states that are as uniform as possible over the state space. Training the goal-conditioned policy is then performed by leveraging the uniform states generated by ChronoGEM as a dataset of goals that provide well-distributed coverage over achievable states. We perform two types of experiments to illustrate C3PO's benefits over similar methods: First, we evaluate entropy upper bounds on ChronoGEM's generated state distribution compared to other reference exploration methods such as RND (Burda et al., 2018) and SMM (Lee et al., 2019) as described in Section 2.1.2. Second, we compare the full C3PO approach to ablated versions that leverage datasets generated by SMM and RND and a random walk. We do this by performing cross-validating of goal-conditioned policies across methods. By training a policy on one method's dataset and evaluating its goal-achievement capabilities on datasets generated by other methods, we can observe which of the methods gives rise to the most general policy. Through these two empirical studies, we illustrate the superiority of ChronoGEM compared to RND and SMM. Finally, we investigate C3PO's abilities in achieving arbitrary poses on five continuous control environments: Hopper, Walker2d, HalfCheetah, Ant and Humanoid. Videos of the resulting behaviours reaching the goal poses are available as gif files in our supplementary material.

## 2 CONDITIONED CONTINUOUS CONTROL POLICY OPTIMIZATION (C3PO)

The optimal universal goal-achievement policy for a given embodiment should allow an agent to achieve any reachable position and pose in the environment as quickly as possible. Learning such a policy necessarily requires a significant amount of exploration and training to both find and learn to attain a large enough number of states to generalize across goal-states. However, in the context of a simulator, which allows for both parallelization and arbitrary environment resets, covering a massive amount of the state space is doable. In our case, we consider $2^{17}$ parallel trajectories, that are re-sampled every step to maintain an high coverage of the reachable space. Once such large coverage of the state space is achieved, learning a goal-achievement policy can be done with a relatively straight-forward learning algorithm that aims at attaining goals from this high-coverage set of states. If the states are well-enough distributed, and if the learning algorithm is sufficiently efficient, we may expect the final policy to achieve universal goal-achievement.

### 2.1 MASSIVELY ENTROPIC PRE-TRAINING

As described above, the first step is to discover the set of achievable goals. This collection will be the key of the effectiveness of the resulting policy: We want it as uniform as possible such that no reachable region is neglected. Therefore, without any prior, the ideal set of goals should be uniformly sampled from the manifold of states that are reachable in a given number of steps ($T$). Since the shape

of that manifold is totally unknown and can be arbitrarily complex, such sampling is impossible. However, it is possible to approximate such a sampling if enough states are simultaneously explored at the previous time step $(T-1)$. Assume we are able to sample $N$ states that approximate the uniform distribution at time $T-1$. Then, from each of these states, playing $K$ uniform actions to obtain $NK$ next states would not lead to a uniform sampling over the possible next states. However, with $N$ large enough, it would at least induce a good coverage of the set of reachable next states. Let $\rho_T$ be the distribution induced by these $NK$ next states. Since the set of achievable states in $T$ steps is necessarily bounded (at least in any realistic environment), and given that we are able to closely estimate $\rho_T$, we can sub-sample with a probability weighted by the inverse of the density $\frac{1}{\rho_T}$ in order to approximate a uniform sampling. We prove in Appendix **??** that such a sub-sampling approximates a uniform sampling when the number of sampled states is large enough. This suggests a recursive approach to approximate a uniform sampling of the reachable states in $T$ steps, by starting with states sampled from the environment's initial distribution $\rho_0$, playing uniform actions, sub-sampling to get an highly covering set that approximates a uniform distribution, re-exploring actions from that new set, and then again, for $T$ iterations. We call this process ChronoGEM (for Chronological Greedy Entropy Maximization) since at a given step, it only focus on maximizing the entropy by directly approximating a uniform distribution over the next step, without further planning. Algo 1 summarizes ChronoGEM.

---

**Algorithm 1** Chronological Greedy Entropy Maximization (ChronoGEM)

---

1: Sample $N$ states $S_0 = \{s_0^i\}_{i=1}^N \sim \rho_0$.
2: **for** $t = 1$ **to** $T$ **do**
3:     Sample $K$ uniform actions for each state of $S_{t-1}$.
4:     Obtain $KN$ next states.
5:     Estimate $\rho_t$ using a density model fitted on the distribution of these $KN$ states.
6:     Sample $N$ states with probabilities $p(s) \propto \frac{1}{\rho_t(s)}$ to get $S_t$.
7: **end for**
8: **return** $S_T$

---

ChronoGEM requires exactly $KNT$ interactions with the environment, which makes easily controllable in term of sample complexity. At each time step, the $N$ sampled states being supposed to be independent also simplifies the implementation, allowing to parallelize $N$ jobs that only consume $KT$ interactions, significantly shortening the time complexity (that also depend on the density model fitting).

### 2.1.1 RESETTABLE STATES ASSUMPTION

Like other diffusion-based algorithms (see related works 4.4), ChronoGEM needs to explore many actions from a single given state. While many approaches would forbid such assumptions to better approximate a real-world situation, we argue that in our case, simulated environments are used for these precise types of simplification, also allowing to run many jobs in parallel and to safely explore dangerous trajectories. In this paper, we ran every simulations using Brax (Freeman et al., 2021), a physics engine offering a collection of continuous control environments similar to MuJoCo (Todorov et al., 2012). Brax is designed to use acceleration devices and massive parallelization, and allows resetting a trajectory at any given state. In the unfortunate case where this assumption would not be available, different equivalent alternative of ChronoGEM could be imagined. With short horizon and small branching factor $K$, the easiest one being to start with $K^T N$ states, and sub-sampling with a factor $K$ at each step, until it ends up with $N$ states at time $T$. In this paper, we stick to the case where this assumption is satisfied.

### 2.1.2 DENSITY ESTIMATION

Many choices of density estimation in high dimensional space exist, from simple Gaussian estimators to neural networks-based methods such as autoregressive models or normalizing flows. The performance of these models may vary given the type of data: some models are more suited for images while others are better for text or lower dimensions. In our case, we wanted to find the most accurate model for the special case of state distribution in continuous control environments.

For this, we implemented 7 candidate models, including Gaussian models (Multivariate, Mixture), autoregressive networks (RNade (Uria et al., 2013), Made (Germain et al., 2015)), and normalizing flows (real-NVP (Dinh et al., 2016), Maf (Papamakarios et al., 2017), NSF (Durkan et al., 2019)). Each of them was trained by maximum likelihood estimation over different continuous control environments, using sets of states obtained from pre-trained agents solving the task. We used various hyper parameters configuration for each model, and selected the best ones based on AUC criteria, then we compared the different models when trained with their best hyper parameters. We found that normalizing flows performed significantly better than other models, and among normalizing flows NSF worked slightly better than other flows. This experiment is detailed in Appendix **??**.

### 2.1.3 CONTINUOUS MAZE

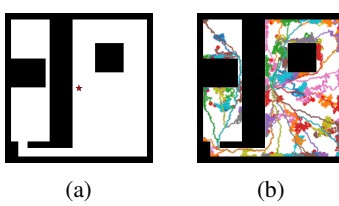

(a)    (b)

Figure 2: (a) Maze environment, the red star corresponds to the starting point and black volumes are walls. (b) Example of ChronoGEM trajectories in the Maze.

As a tool scenario to test ChronoGEM, we implemented a two-dimensional continuous maze in which actions are $d_x, d_y$ steps bounded by the $[-1, 1]^2$ square, and the whole state space is bounded by the $[-100, 100]^2$ square. This environment, illustrated in Figure 2a is similar to the maze used by Kamienny et al. (2021), except it adds a significant difficulty induced by the presence of two narrow corridors that needs to be crossed in order to reach the top-left room. The main goal of this experiment is to verify that even in a challenging tool game, ChronoGEM still manages to induce a uniform distribution over the whole state space. In order to emphasize the relative difficulty of the exploration of this maze, we also run a random walk, SMM and RND to compare the obtained state coverages. We implemented ChronoGEM with $N = 2^{17}$ paralleled environments and branching factor $K = 4$. In this setup we know that if T is large enough, all achievable states are just every point in the maze, so ChronoGEM could be uniform on the maze given that we let it run for enough time (for instance, $T = 1000$). However, in that given episode length, both RND and SMM failed at exploring beyond the first corridor, and a random walk did not even explore the whole first room, as illustrated in Figure 3.

### 2.2 GOAL-CONDITIONED TRAINING

To build C3PO, we modified Brax' implementation of SAC to take a set of goal as input and train to reach them. The reward is the opposite of the maximum of the euclidean distance between a body (e.g. an arm, the torso, a leg) and its goal position. As a result the policy is encouraged to move to the correct location and then match the correct pose. The goal is added to the observation as a relative position to the current position. we say that a goal is reached if the Euclidean distance between the agent's state and the goal is smaller that a tolerance threshold $\epsilon$. In other terms, an episode $\mathcal{E}$ is successful when its closest state to the goal is close enough: $\text{success}(\mathcal{E}|g) \Leftrightarrow \min_{s \in \mathcal{E}} ||s - g||^2 < \epsilon$. We set the environment to stop an episode as soon as it is successful, or when it exceeds the number of allowed steps. We initially set the tolerance $\epsilon$ to be high (1.) and slowly anneal it down when the success rate reaches 90% on the training data. As a result SAC learns first to move towards the

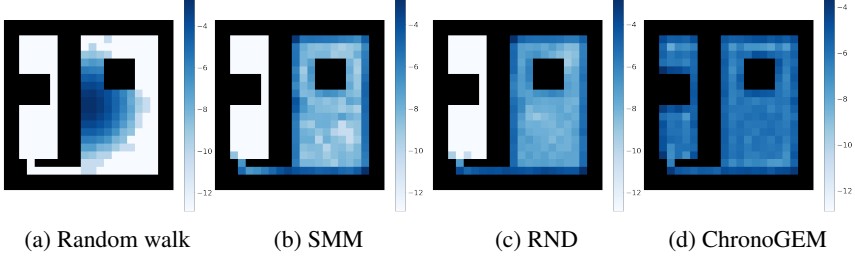

(a) Random walk    (b) SMM    (c) RND    (d) ChronoGEM

Figure 3: Log-frequencies of the discretized states visitation when taking the last states from 4000 episodes, sampled according to a random walk (a), SMM (b), RND (c) and ChronoGEM (d), averaged over 10 seeds. Only ChronoGEM managed to visit states in the top-left room.

target and then to match the exact position. We call C3PO the resulting procedure that combines ChronoGEM for the training data collection and goal-conditioned SAC with tolerance annealing, as described in Algorithm 2.

---

**Algorithm 2** C3PO

---

**Require:** Initial tolerance $\epsilon$.
 1: Collect training goals G with ChronoGEM.
 2: **for** enough steps **do**
 3:     Draw goals from G.
 4:     Run full episode rollout with drawn goals, considering an episode is successful if the distance to the goal is less than $\epsilon$.
 5:     Add rollouts to the replay buffer.
 6:     If the success rate is above 90%, multiply $\epsilon$ by .99
 7:     Train networks using SAC losses on transitions drawn from the replay buffer.
 8: **end for**
 9: **return**  Trained policy.

---

## 3 Experiments

This section has two functions: 1) to quantify the superiority of C3PO compared to baselines in which the training data was collected using different exploration methods (SMM, RND and a random walk) and 2) to illustrate the accuracy of the resulting goal-achieving policy after reaching asymptotical training performances, on various continuous control tasks, including Humanoid. To collect training data, ChronoGEM was run with $N = 2^{17}$ paralleled environments and branching factor $K = 4$ in all following experiments, except for Humanoid in which $N = 2^{15}$ and $K = 64$. We detail C3PO and all baselines (SAC+RND, SAC+SMM and SAC+random walk) implementations in Appendix **??**. For each baseline, we separately tuned the hyper parameters in order to obtain the best performance in all different environments.

### 3.1 Continuous control tasks

We used the control tasks from Brax (Freeman et al., 2021) as high dimensional environments. To compare the entropy and the richness of the obtained set of states with bonus-based exploration baselines, we focused on four classical tasks: Hopper, Walker2d, Halfcheetah and Ant. Since we focus on achieving arbitrary poses and positions of the embodied agents, we modified the environments observations so they contain the $(x, y, z)$ positions of all body parts. All measures (cross entropy in section 3.2 and reaching distances in section 3.3) are based on that type of state. To get reasonable trajectories (as opposed to trajectories where HalfCheetah jumps to the sky), we explore the environment within the low energy regime by putting a multiplier on the maximum action. The multiplier is .1 for Hopper, .1 for Walker, .01 for HalfCheetah and 1. for Ant. In the two following subsections, we considered episodes of length $T = 128$. So the physical time horizon is similar on all tasks, we added an action repeat of 6 for Hopper and Walker. All episode end conditions (because the torso is too low or too high for example) have been removed, so we have no prior.

### 3.2 Entropy upper-bound

Given a set of points $x_1 \dots x_N$ sampled from a distribution with an unknown density $\rho$, one can estimate an upper bound of the entropy of $\rho$ given by the cross entropy $H(\rho, \hat{\rho})$ where $\hat{\rho}$ is an estimation of $\rho$:

$$H(\rho, \hat{\rho}) = -\mathbb{E}_{x \sim \rho}[\log \hat{\rho}(x)] = H(\rho) + \mathrm{KL}(\rho || \hat{\rho}) \geq H(\rho).$$

The estimation $\hat{\rho}$ being trained by maximum likelihood specifically on the set of points, it directly minimises the cross entropy and closely approximate the true entropy. The KL term becomes negligible and only depends on the accuracy of the trained model on the observed set of points, which supposedly does not differ given the different exploration method that generated the points. Consequently, comparing the cross entropy is similar to comparing the entropy of the distribution

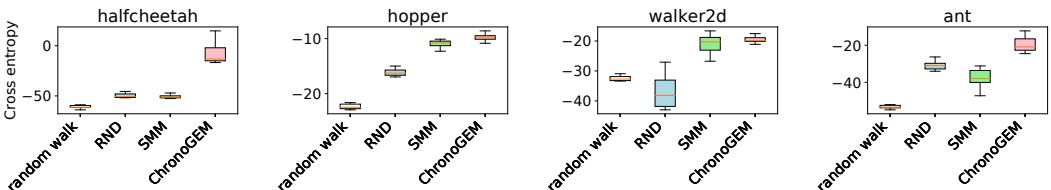

Figure 4: Distribution over 10 seeds of the cross entropies of the state visitation induced by Chrono-GEM, RND, SMM and a random walk, on different continuous control tasks.

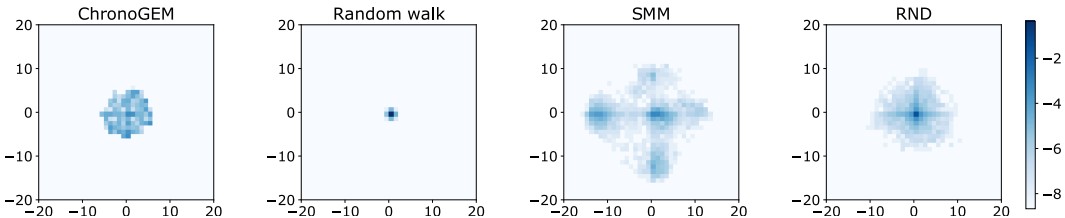

Figure 5: Sky-view of the discretised spatial log-frequency covered by Ant with the different exploration methods. SMM has the largest scope but contains empty zones even close to the origin. Both RND and ChronoGEM share similar range covering all directions, and ChronoGEM is visibly much more uniform, while other methods are concentrated on the origin. Note that this only represents the spatial positions, while both poses and positions are being explored.

induced by the exploration. In this experiment, we used this upper-bound to compare the efficiency of ChronoGEM compared to RND, SMM and a random walks. Figure 4 displays box plots over 10 seeds of the resulting cross entropy measured on the sets of states induced by different algorithms, on the 4 continuous control tasks. As expected, the random walk has the lowest entropy, and RND and SMM have, in average over the environments, similar performances. ChronoGEM has the highest entropy on all environments, especially on HalfCheetah, where it was the only method to manage exploration while the actions were drastically reduced by the low multiplier (see 3.1). In order to illustrate the fact that ChronoGEM induces a distribution that is close to the uniform, we measured the spatial coverage based on a discrete gird of the x-y plan: if the distribution is uniform over both the possible poses and positions, it should be in particular uniform over the positions. Figure 5 shows the resulting log-frequency on the x-y grid visitations and if ChronoGEM is not the method that induces the largest scope of exploration, it however has the most uniform coverage. We also report in appendix **??** the x grid visitation in 2D environments (Hopper, Walker2d and Halfcheetah).

## 3.3 COMPARISON OF EXPLORATION METHODS VIA GOAL-CONDITIONED TRAINING.

If an exploration method is good, drawing from the states it explored should be a good approximation of drawing from all achievable states. The state distribution induced by an exploration algorithm can be used both as a training set of goal, but also as an evaluation set of goals. In the next experiment, for each environment, we ran every of the four examined exploration methods (ChronoGEM, Random Walk, SMM and RND) with 3 seeds to build 3 training goal sets per method and 3 evaluation goal sets per method. Training goal sets have 4096 goals and evaluation goal sets have 128 goals. We plot the success rate with regard to the tolerance, for each environment and evaluation goal set. Figure 6 shows that evaluated on ChronoGEM goals, only C3PO – which is trained on ChronoGEM – gets good results while evaluated on goals from other methods. This is a good hint that the diversity of ChronoGEM goals is higher than other exploration methods. C3PO performs well on other evaluation sets as well, in particular in the low distance threshold regime (see Hopper and Walker). This can be explained by the fact that C3PO learns to reach a high variety of poses, since being able to achieve poses with high fidelity is what matters for low distance threshold regime. However, these achievement rates alone are still hardly interpretable: for example, being good at reaching goals generated by the random walk is less important than achieving harder goals, especially those from the highly entropic distributions (like ChronoGEM goals on Halfcheetah or SMM goals on Walker). We hence summarized the results by collecting all the areas under the curve (AUC), and weighting

them proportionally to the exponential of the evaluation goals entropy in Figure 7. Indeed, if a set is very diverse, it means more to be able to achieve all of its goals, and vice-versa: if a set is not diverse we don't want to give too much importance to it, as achieving always the same goal is not so interesting. The exponential of the entropy quantifies the number of states in the distribution. We call this metric Entropy Weighted Goal Achievement (EWGA):

$$EWGA(method) = \frac{\sum_{s \in evaluation\ sets} exp(entropy(s)) * AUC(method\ on\ s)}{\sum_{s \in evaluation\ sets} exp(entropy(s))}$$

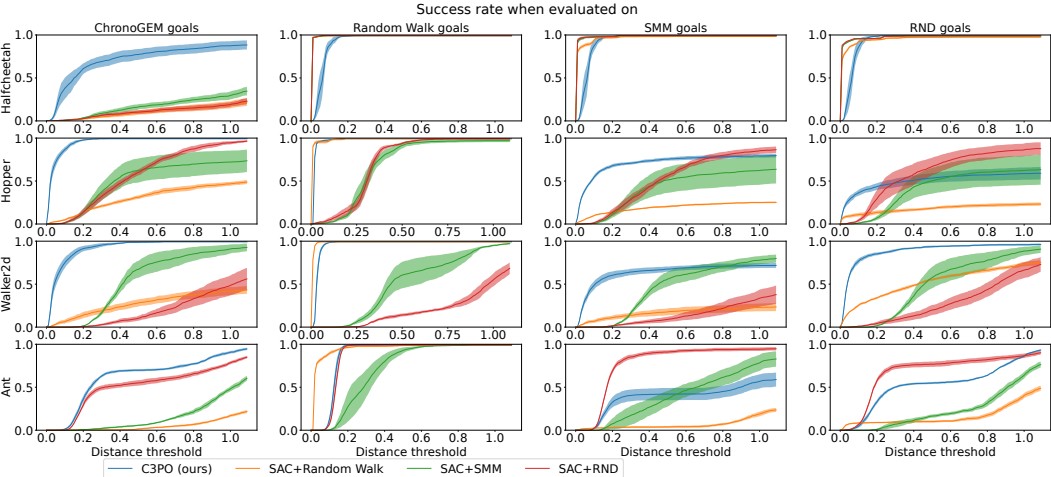

Figure 6: For each environment (lines) and each set of evaluation goals (columns), success rates as a function of distance thresholds obtained by SAC when trained on the different sets of training goals (ChronoGEM, Random Walk, SMM, RND). Each exploration algorithm was run over 3 seeds to collect evaluation and training goals, and each SAC training was also run over 3 seeds, so the resulting curves are actually averaging 9 different values.

### 3.4 MASSIVE GOAL-CONDITIONED TRAINING.

Now that we established that ChronoGEM is the best exploration method for the purpose of producing training goals for a goal-conditioned setup, we will only use this method. We know allow ourselves to train for massive amount of steps, and see what is the best policy we can achieve. Thanks to Brax's high parallelization and efficient infrastructure, it is possible to run 30G steps in a couple days. We also add Humanoid to our set environments. By default, ChronoGEM would mostly explore positions where the humanoid is on the floor. However, it was simple to modulate the algorithm to only explore uniformly in the space of state where the humanoid is standing. For example, on can just associate

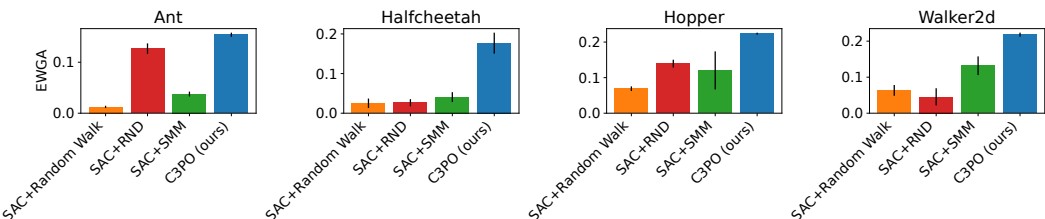

Figure 7: Entropy Weighted Goal-Achievement (EWGA). This estimates the ability of a policy to achieve goal sets that better covers the space (for example, a policy like C3PO that reaches a larger variety of states has an higher EWGA than a policy like SAC trained on the Random Walk, that only reaches states that are close to the origin).

zero weight to undesired states during the re-sampling step. That way, we avoided states in which the torso go under the altitude of .8 (the default failure condition). ChronoGEM is modified to not draw states where the humanoid is too low. The goal-conditioned learner gets a high penalty for going too low too. The visual results of a policy able to achieve 90% success at .25 tolerance are represented in Figure 8. This shows that when we do have a prior, we can leverage it to steer the exploration and policy learning.

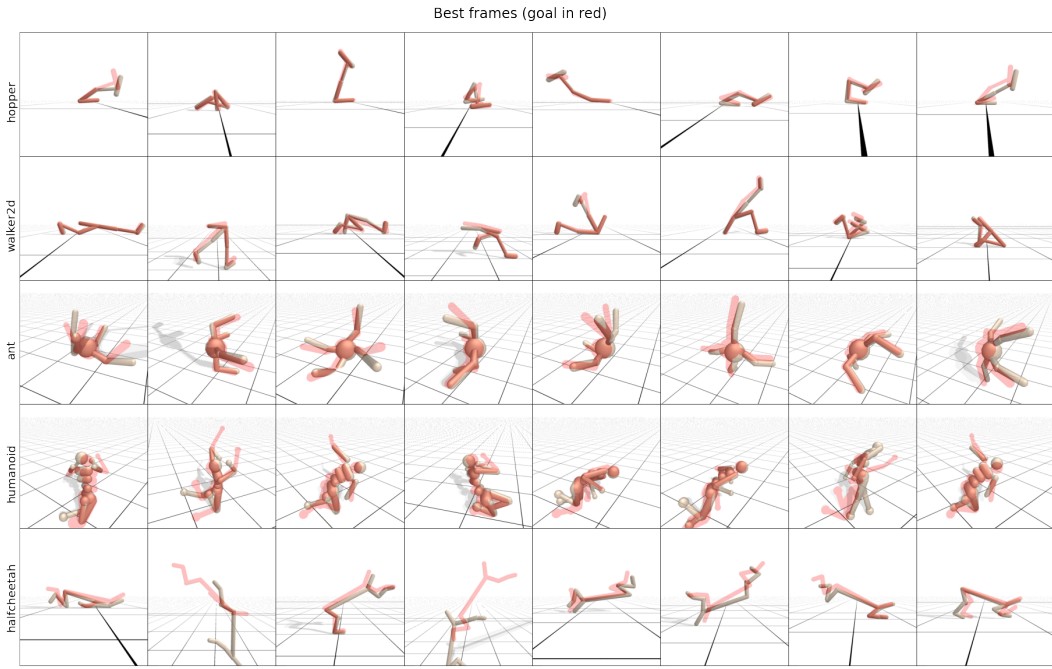

Figure 8: For each environment, we drew 8 goals from the ChronoGEM distribution and ran the policy trained on ChronoGEM goals. This figure represent the frame that is the closest from the goal for each episode, overlayed with the goal in red.

## 4  RELATED WORKS

This work is situated between various fields. Although effectively a goal-conditioned policy optimization algorithm, C3PO is enabled by the ChronoGEM exploration algorithm. We will first look at similar goal-conditioned learning setups, and then discuss more in-depth related works in the domain of exploration.

### 4.1  GOAL-CONDITIONED REINFORCEMENT LEARNING

Goal-conditioned RL (Kaelbling, 1993; Schaul et al., 2015) is the general setup of learning a goal-conditioned policy instead of a specialized policy. We are particularly interested in goal-based setups where there is no *a-priori* reward function. Although well known works such as HER (Andrychowicz et al., 2017) demonstrate methods for learning goal-conditioned policies with minimal explicit exploration, more recent works (Pitis et al., 2020; OpenAI et al., 2021; Mendonca et al., 2021) demonstrate the importance of having a good curriculum of goals to train from. MEGA (Pitis et al., 2020) extends HER-style relabeling and answers the exploration problem by iteratively sampling goals according to a learnt density model of previous goals. ABC (OpenAI et al., 2021) demonstrates the importance of an adversarial curriculum for learning more complex goal-conditioned tasks, but is concentrated on specific tasks instead of arbitrary goal achievemnt. LEXA (Mendonca et al., 2021) builds on Plan2Explore (Sekar et al., 2020), and demonstrates the importance both of a good exploration mechanism, as well as the use of significant amounts of (imagined) data for learning an arbitrary goal-achievement policy. DIAYN (Eysenbach et al., 2018) uses a two-part mechanism that encourages the agent to explore novel areas for a given latent goal, while at the same time learning a

goal embeddings for different areas of the state space. While some of the above methods consider notions of density for exploration (Eysenbach et al., 2018), C3PO uses a more principled exploration mechanism, and is particularly interested in high-precision goal-achievement from full states.

## 4.2 BONUS-BASED EXPLORATION

Although generally not concerned with goal-conditioned RL, there is a large family of exploration methods that are manifest as reward bonuses, with the intent of training a policy faster, or to be more performant. One family of approaches uses state-visitation counts that can be approximate to create an associated bonus for rarely-visited states (Bellemare et al., 2016; Ostrovski et al., 2017). Prediction-error bonuses use the residual error on predictions of future states as a reward signal to approximate novel states, these includes methods such as RND (Burda et al., 2018) which leverages the prediction error of random projections of future states, SPR (Schwarzer et al., 2020) and BYOL-Explore (Guo et al., 2022), which make use of the self-prediction error of the network with a frozen version of itself. Model-based methods often optimise for next-state novelty, either by looking at the KL-Divergence between sampled states and likely states, such as in VIME (Houthooft et al., 2016) or by explicitly driving towards states with high model ensemble disagreement such as in Plan2Explore (Sekar et al., 2020). RIDE (Raileanu and Rocktäschel, 2020) and NGU (Badia et al., 2020) use episodic memory in which the bonus reflects the variety of different states covered in a single trajectory.

## 4.3 ENTROPY MAXIMISATION

Some exploration algorithms, such as ChronoGEM, are constructed in order to maximize the entropy of the state visitation distribution. Most of them however, focus on the distribution induced by the whole history buffer (instead of the just $T$-th states of episodes in ChronoGEM), generally based on the behavior of a trained policy. This is the case of MaxEnt (Hazan et al., 2019), GEM (Guo et al., 2021), SMM (Lee et al., 2019) and CURL (Geist et al., 2021). In APT (Liu and Abbeel, 2021b), instead of using a density model to estimate the entropy, they use a non-parametric approach based on the distance with the K nearest neighbors in a latent representation of the state space. APS (Liu and Abbeel, 2021a) combines APT's entropy bonus with an estimation of the cross-entropy based on successor features to maximize the mutual information $I(w; s)$ between a latent skill representations $w$ and states.

## 4.4 DIFFUSION-BASED EXPLORATION

ChronoGEM is based on a tree-structured diffusion, that makes a selection of states, randomly explore from these states and then reselect states, etc. Go-Explore (Ecoffet et al., 2019), share the same approach, by running a random policy for some steps, then selecting a set of 'interesting' states, then going back in these states and start again random explorations from them. The main difference with ChronoGEM is that we skip the 'go back' part and we only perform one step of random actions before the new state selection. Also, the selection of states in ChronoGEM is provably converging to a uniform distribution over achievable goals, and does not need any additive prior about the state importance. Another close work also using a diffusion approach is UPSIDE (Kamienny et al., 2021). It finds a set of nodes along with a set of policies that connect any node to the closest ones, looks for new nodes by random exploration from the existing ones, and remove non necessary nodes that are reached by the less discriminable policies. UPSIDE converges to a network of nodes that efficiently covers the state space.

## 5 CONCLUSION

We designed ChronoGEM, an exploration method that generates high entropy behaviors, in theory (Theorem **??**) and in practice (Figure 4), outperforming baseline algorithms. All the skills discovered by an exploration algorithm can be used to train a goal-conditioned policy. We showed that training ChronoGEM goals results in the most potent policies compared to other exploration methods. On Hopper, Walker, HalfCheetah, Ant and Humanoid, visuals and metrics show that the policy we trained is able to achieve a large variety of goals - by moving to the correct position and then matching the pose - with high fidelity.

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
