# OpenReview forum: "C3PO: Learning to Achieve Arbitrary Goals via Massively Entropic Pretraining"
_ICLR.cc/2023/Conference — Submitted to ICLR 2023_

### Official Review · Reviewer_4RNg · 2022-10-18

**Confidence:** 4
**Correctness:** 3
**Technical Novelty And Significance:** 2
**Empirical Novelty And Significance:** 3
**Recommendation:** 3

**Clarity, Quality, Novelty And Reproducibility:**

The paper is mostly clear, of OK quality, incrementally novel, and seemingly reproducible provided sufficient computational resources.

**Strength And Weaknesses:**

## Strengths

**Clarity:**

- The figures are, for the most part, very nice and clear.

**Experiments/Results:**

- ChronoGEM does work well as an exploration procedure, and GIFs in the supplementary demonstrate that their goal-conditioned policy can reach many arbitrary poses.
- C3PO is able to do well on goal distributions generated by other methods, verifying the fact that ChronoGEM is a good exploration algorithm.

## Weaknesses

**Clarity:**

- Quite a few grammatical hiccups throughout. Please fix (some that I found are listed in the minor details heading)

****************Framing:****************

- I somewhat doubt the utility of this work. It requires a specific setting (a cheap simulator that can be sampled en masse in parallel, and reset to arbitrary states), and has extremely poor sample efficiency due to this assumption. Essentially, ChronoGEM tries to visit all of the states in the simulator. In anything other than locomotion tasks, (e.g. robotic manipulation) these states will mostly be useless and having so many states to try to achieve can hurt learning.
- What is the point of being able to achieve arbitrary positions and poses? The authors state in the abstract that it would allow for easier control and be re-usable as a key building block for downstream tasks. But they don’t show this. The experiments simply demonstrate that Goal-conditioned SAC can reach the position and poses given.
- Conclusion: “In the real world, no reward function is provided.” THis is true, however in the real world we also can’t run ChronoGEM. This goes back to how I think the paper is not framed too well.

**Experiments:**

- Related to the above point about framing in achieving arbitrary positions and poses, the authors should show experiments where their method allows for better finetuning performance on some downstream tasks (maybe goal conditioned, maybe not) to validate the claims made in the abstract and intro.

****************************Minor details:****************************

- Grammar
    - Intro last paragraph: “its’ → its”
    - Sec 2.1: “necessary bounded” → “necessarily bounded”
    - “up-left room” → “top-left room”
    - “let it enough time” → “let it run for enough time”
    - “we says” → “we say”
- Questions
    - In chronoGEM, how exactly do you sample the ****next**** states? Do you just reset BRAX simulation to those exact states and continue from there?

**********************Formatting:**********************

- The formatting is incorrect. The margins are way too small. This is unfair to other papers who fit everything into the page limit without the small margins. Regardless of whether this was intentional or on accident, I am giving a strong reject no matter what, until the authors fix this issue during the rebuttal.

**Summary Of The Paper:**

The authors present a method that performs massive uniform exploration of simulated environments and then trains goal-conditioned SAC with an annealed success condition to achieve states discovered during exploration. They demonstrate this exploration procedure to be superior on locomotion environments to other methods.

**Summary Of The Review:**

While some of the results are nice, I have 3 main issues with this paper, as detailed above: 1) Framing, 2) The formatting, and 3) The experiments.

Each of these is important and the formatting is critical (there are many papers already desk-rejected for incorrect formatting), and as such, I cannot recommend anything better than a strong reject.

---

> ### Author Response · Authors · 2022-11-18
> **Author response**
>
> << I somewhat doubt the utility of this work.
>
> While the paper focuses on having as little prior as possible, for humanoid we did add a prior that only poses that are not too low are interesting. As a result, for those poses we could potentially imagine a sport or a dance where they are interesting. But if the user has extra prior knowledge, they would be free to add them in to reduce the complexity of the task - and therefore the sample complexity.
> Even in robotic manipulation, one may want to achieve all possible settings.
> We would like to note that this work is to our knowledge the first successful application of unsupervised continuous control of humanoid in high dimension (copying the position and the pose)
>
> << What is the point of being able to achieve arbitrary positions and poses?
>
> That’s a good point. If someone wants to learn how to dance, they will probably watch experienced dancers, what position and poses they achieve, and how. A robot cannot do that, but the result of our paper provides a building block for a possible solution, but this is beyond the scope of the paper.
> A large part of tasks in RL literature is about reaching states. In continuous control, states could be the poses and positions, it could also be joints angles, speed and accelerations. All these choices of what is a state have advantages and disadvantages. Reaching positions forces the agent to learn how to move in any direction, and reaching the pose forces a fine control of all the joints.
>
> << Conclusion: “In the real world, no reward function is provided.” This is true, however in the real world we also can’t run ChronoGEM. This goes back to how I think the paper is not framed too well.
>
> Indeed, we’re going to change this claim to make it match the paper spirit better.
>
> << Related to the above point about framing in achieving arbitrary positions and poses, the authors should show experiments where their method allows for better finetuning performance on some downstream tasks (maybe goal conditioned, maybe not) to validate the claims made in the abstract and intro.
>
> Reaching a new unseen pose and position is already a specific downstream task. Which is solved in a zero-shot application of the learned policy.
>
> << Minor typos
> Thanks for finding them, we added the correction in the revision.
>
> << In chronoGEM, how exactly do you sample the next states? Do you just reset BRAX simulation to those exact states and continue from there?
>
> Yes, BRAX allows reseting episodes at the selected states.
>
> << The formatting is incorrect.
>
> This was a latex accident (an unfortunate mistake due to a package import), sorry about this and thanks for bringing this up.
> We corrected the format in our revision, please note that it did not affect the page count as while we did not remove anything (except for other suggested revisions).

---

> > ### Comment · Reviewer_4RNg · 2022-11-18
> > **Response to author comments**
> >
> > Thanks for the response. Some more comments below:
> >
> > > Reaching a new unseen pose and position is already a specific downstream task. Which is solved in a zero-shot application of the learned policy.
> >
> > This is fair, although reaching poses is not a very interesting downstream application as that is exactly what the policy was trained to do, and with very little difference between training and testing task distributions. The authors could consider tasks such as running, jumping, imitating certain trajectories, etc. as more interesting downstream fine-tuning applications to prove the claims made.
> >
> >
> >
> > In general, after the new formatting and responses, I am raising my score to a 3. I'm sure this is a bit disappointing to hear that it's still a reject, but this is because the authors made very few changes to the actual paper during the 2 week rebuttal period. After looking at the other reviews and the author responses, it seems that the authors are promising many changes but given that they did not implement them in time, I believe the paper would be better off being submitted to the next conference with some framing and writing changes, additional experiments, and one extra comparison to Skew-fit as mentioned by other reviewers (although I do believe that ChronoGEM will work better than Skewfit).

---

### Official Review · Reviewer_3jQz · 2022-10-21

**Confidence:** 3
**Correctness:** 3
**Technical Novelty And Significance:** 2
**Empirical Novelty And Significance:** 3
**Recommendation:** 3

**Clarity, Quality, Novelty And Reproducibility:**

The paper is generally clearly written, and experiments are conducted thoroughly with multiple random seeds. The method is described with sufficient implementation details that it seems likely to be reproducible.

Questions:
- Given that the entropy needed to be upper bounded in section 3.2, how is the entropy used for the computation of the EWGA metric?
- Is the entire observation of each environment replaced by the xyz positions, or are they appended to the default state spaces?

Other comments/nitpicks/typos:
- HalfCheetah has inconsistent capitalization throughout the manuscript
- I think the first sentence in Section 3.4 is overstated: while it does seem that ChronoGEM performs the best for the studied environments, the statement is rather general and this hasn’t been shown for all goal-conditioned setups.


**Strength And Weaknesses:**

Strengths:
- The ChronoGEM algorithm appears empirically to generate more uniform state distributions compared to random walks, SMM, and RND exploration strategies on the maze and ant environments.  It is also conceptually simple to understand and implement.
The experimental results training goal-conditioned RL on the goals collected by ChromoGEN outperforms policies trained on the other goal distributions, which is a positive signal that the distribution of goals for ChromoGEM is relatively wide.

Weaknesses:
- The theoretical argument in section 2.1 does not seem convincing to me. The statement seems to be that the state visitation will eventually converge to be uniform over the entire state space, but the theorem in Appendix A only demonstrates that the states selected for the next step of the greedy procedure can be approximately uniform from KN sampled states. As the authors mention, playing uniform actions would not necessarily lead to a uniform sampling over the possible next states. So it seems like the “inductive” step here is missing. So I don’t agree with the statement in the conclusion that the method generates “high entropy behaviors, in theory (Theorem 1)”.
- I think it would help the paper to have a comparison to a method like Skew-Fit[1] that has a similar exploration strategy to the one proposed in the paper by rebalancing the distribution of states to set as goals.
- There isn’t an ablation study conducted on the impact of the tolerance annealing component from SAC to see how much of the improved performance comes from that compared to the goals from ChronoGEM.
- The EWGA metric is an interesting way to evaluate the performance across evaluation sets that may have varying difficulties, but why not evaluate using another strategy like uniformly sampling goals across the [x, y, z] space? That seems like it would be less noisy, and that observation space has already been constructed.

[1] Skew-Fit: State-Covering Self-Supervised Reinforcement Learning (Pong et. al, 2019)

**Summary Of The Paper:**

This paper presents a method called Entropy-Based Conditioned Continuous Control Policy Optimization (C3PO) that tackles the problem of learning a general goal-conditioned policy in two stages. In the first stage, using an exploration algorithm called ChronoGEM, an exploration dataset consisting of diverse goals is collected. In the second stage, the goals collected in the first phase are used as goal targets in goal-conditioned RL with soft-actor critic (SAC) as the base algorithm. The paper demonstrates the method’s effectiveness on Gym environments: walker, hopper, halfcheetah, and ant. It demonstrates that C3PO is not only able to reach a wider distribution of states compared to RND and SMM, training goal-conditioned policies on these states improves the ability of these policies to solve a range of goals.

**Summary Of The Review:**

The proposed method has promising empirical results, but I find the current version of the theoretical justification unconvincing and the analysis of why the method works well could be more thorough. Therefore I do not think the paper should be accepted in its current form, but if the authors can address many of my concerns above, that would be very helpful.

I also want to leave a note that this paper seems to be improperly formatted: the margins are much smaller than in the original template, the ICLR header is missing, and the title / author list formatting is nonstandard.

---

> ### Author Response · Authors · 2022-11-18
> **Author response**
>
> << The theoretical argument in section 2.1 does not seem convincing to me.
>
> With infinite sampling budget, we show that we'll be uniform over the set of achievable states in K steps at EACH step K of the algorithm. So there is no need for an induction.
>
> << I think it would help the paper to have a comparison to a method like Skew-Fit[1]
>
> Thanks for that related paper, we were not aware of it and will add it to our related work section. The main difference with our approach is that we separated the goal discovery phase to the goal reaching training. We also used a different resampling strategy, directly using the inverse of an estimator, which is possible for us since we are not policy-based during exploration.
>
> << There isn’t an ablation study conducted on the impact of the tolerance annealing component from SAC to see how much of the improved performance comes from that compared to the goals from ChronoGEM.
>
> All comparisons are run with tolerance annealing on each dataset. So the advantage can only come from ChronoGEM.
>
> << why not evaluate using another strategy like uniformly sampling goals across the [x, y, z] space?
>
> If we draw uniformly, we’re almost sure to draw a goal that is impossible to reach (for example one leg of the ant 10 units north and the other 10 units south, with a 2 unit long ant). So we believe evaluating on such a dataset isn’t very representative of quality.
>
> << Given that the entropy needed to be upper bounded in section 3.2, how is the entropy used for the computation of the EWGA metric?
>
> We use the upper bound as well - we don’t believe there is any better solution.
>
> <<Is the entire observation of each environment replaced by the xyz positions, or are they appended to the default state spaces?
>
> The observation space is a concatenation of the xyz positions of bones + the default angle/speed observations.
>
> << HalfCheetah has inconsistent capitalization throughout the manuscript
>
> Thanks for noticing, we will make it consistent.
>
> << I think the first sentence in Section 3.4 is overstated: while it does seem that ChronoGEM performs the best for the studied environments, the statement is rather general and this hasn’t been shown for all goal-conditioned setups.
>
> We meant that ChronoGEM is the best compared to our baselines. We will correct the formulation.
>
> << the margins are much smaller than in the original template, the ICLR header is missing, and the title / author list formatting is nonstandard.
>
> Formatting was an unfortunate mistake due to a package import.

---

> > ### Comment · Reviewer_3jQz · 2022-11-27
> > **Response to author comments**
> >
> > Thank you for your detailed response. Thank you for clarifying my initial comment about the inductive step for the theoretical argument in section 2.1; in hindsight my comment may not have been technically accurate. However, I still feel that making this argument *in the limit of infinite samples* does not really help justify why the algorithm practically works well, so I feel that the importance of the theorem is slightly overstated in the manuscript. I also echo the concerns of other reviewers about the sampling and simulator requirements of the setting considered, and at the time will keep my score.

---

### Official Review · Reviewer_nd1d · 2022-10-24

**Confidence:** 4
**Correctness:** 2
**Technical Novelty And Significance:** 2
**Empirical Novelty And Significance:** 3
**Recommendation:** 3

**Clarity, Quality, Novelty And Reproducibility:**

Additional Comments on the paper:

- It's not discussed very clearly in the paper, but there are obvious practical challenges in being able to create a policy that can reach all of the states in a very uniform distribution. There will be more probability for states that are closer to the initial State distribution for the trained RL policy.
- What are the assumptions that go along with the proposed algorithm in section 2.1? From this somewhat difficult-to-understand analysis it seems that one additional assumption is that the environment is deterministic and can reach these states given similar paths and actions. Generally, the assumption and motivation that these methods will work really well in simulation can be a problematic hypothesis and can't imply that this method will not be helpful on any real-world problems that have data limitations, stochasticity and partial observation.
- It's not clear what is the goal of algorithm one. The first sentence after that algorithm states that it only requires NKT samples, but it doesn't say what these are required for.
- The stopping condition for individual episodes that depends on $$\epsilon$$, how was $$\epsilon$$ chosen for each of the environments? In particular, how was it chosen such that it does not bias the method proposed in the paper compared to other methods?
- The appendix links in the paper appear to be broken.
- Why are the XYZ positions of everybody part added to the state in the simulation and not just the position information for the center of the agent? For example, the center of mass could be used or the agent's position of the root link.
- The beginning of section 3.2 on entropy upper bounds needs citations to inform the reader where these axioms are derived from.
- The results in figure 4 that compare different methods depending on the measure proposed for entropy over a prior distribution appear to be somewhat self-fulfilling. The method proposed in this paper may be optimizing almost this exact objective, while prior methods are not. how can we be sure that this is the most critical form of the objective to be optimized and that the other methods aren't performing much better, giving some other analysis of the entropy over the state space?
- Why is this method not compared to a PT and APS? both of those methods perform types of entropy maximization to explore states in the environment. this makes them appear to be very likely candidates for solving a similar problem.
- In addition, this method should also be compared to the UPSIDE algorithm.

Additional works that should be cited in the paper:
- Skew-Fit: State-Covering Self-Supervised Reinforcement Learning, Vitchyr H. Pong, Murtaza Dalal, Steven Lin, Ashvin Nair, Shikhar Bahl, Sergey Levine

**Strength And Weaknesses:**

pros
- The paper proposes finding a more helpful policy that can be trained inside the environment compared to prior methods. Most prior curiosity-based methods do help the agents learn to discover different states that are more difficult to reach via exploration bonuses, but this method also seems to train a goal condition policy making it more taskable as well as prior methods.
- They are also able to show through some illustrative examples that their proposed method can or at least appears to explore the environment in a more uniform manner.

cons
- There are some limitations in the comparison and analysis that should be addressed. In particular, some prior methods are mentioned in the paper that would be important to compare to understand better if the proposed method is better than prior work. this also notes and is based on the concept that they're training a goal condition policy to explore the state distribution and not just a curiosity-based method.
- This method appears only to be applicable to simulators. As the authors note, it's also extremely data-hungry.

**Summary Of The Paper:**

this paper is proposing a new unsupervised reinforcement learning scheme for being able to train a more taskable agent that can reach a large base of goals in the environment. the motivation for this work is to not only collect the data and have the agent be able to explore but at the same time be able to train a goal condition to policy such that if we wanted the agents to reach a particular state in the world, we could specify that during test time. the method proposes to do a type of incremental search outward from the initial State distribution and slowly add states that the agent can learn how to reach in this case in an effort to be able to sample from the state distribution such that goals can be reached uniformly across the state space. the method does perform some analysis showing that qualitatively the agents appear to learn and display a more uniform type of visitation strategy across the state space while visiting goals. and the method is compared across a handful of mujoco-based robotic simulations.

**Summary Of The Review:**

The paper proposes a helpful algorithm to allow for training a goal-based agent to reach many goals. The work requires further details and comparisons to prior methods to understand if it is an improvement over prior work.

---

> ### Author Response · Authors · 2022-11-18
> **Author response**
>
> << This method appears only to be applicable to simulators. As the authors note, it's also extremely data-hungry.
>
> Many problems can be simulated. It is indeed data hungry depending on the task, but we show the result is a universal goal achieving policy. So instead of training once per task, we train once for all tasks, which is a small extra cost to pay for this result.
>
> << It's not discussed very clearly in the paper, but there are obvious practical challenges in being able to create a policy that can reach all of the states in a very uniform distribution (...)
>
> The ChronoGEM algorithm is designed to compensate for this problem by sampling proportionally to 1/probability. See section 2.1. We can see in Fig. 3 that when other exploration methods do suffer from the issue you’re raising, ChronoGEM does produce a uniform distribution.
> The policy that is learned by C3PO is a goal conditioned policy, not a policy that reaches goal uniformly.
>
> << What are the assumptions that go along with the proposed algorithm in section 2.1? (...)
>
> ChronoGEM (algorithm 1) is an exploration algorithm, and doesn’t need any of the assumptions you mentioned to function - actually it uses random actions, so it is stochastic and adding stochasticity to the environment is not a problem.
> Stochastic environment makes goal conditioned tasks more difficult to achieve, so the second step where we learn to achieve discovered goals would be more challenging, but in principle entirely possible.
>
> << It's not clear what is the goal of algorithm one. (...)
>
> Algorithm 1 discovers achievable goals.
>
> << The stopping condition for individual episodes that depends on ϵ, how was ϵ chosen for each of the environments? (...)
>
> This is explained in section 2.2: We initially set the tolerance ϵ to be high (1.) and slowly anneal it down when the success rate reaches 90% on the training data.
> Since the difficulty adapts to the success on the training data, we don’t feel that it advantages any method.
>
> << The appendix links in the paper appear to be broken.
>
> You can find the version with the appendix in the supplementary materials.
>
> << Why are the XYZ positions of everybody part added to the state in the simulation and not just the position information for the center of the agent?
>
> Then the task would be extremely easy. We want to agent to match the position AND the pose. Also using the center of mass is a prior on the task we want to achieve - and we want to have as few priors as possible.
>
> << The beginning of section 3.2 on entropy upper bounds needs citations to inform the reader where these axioms are derived from.
>
> This is by definition of the cross entropy https://en.wikipedia.org/wiki/Cross_entropy.
>
> << The results in figure 4 that compare different methods depending on the measure proposed for entropy over a prior distribution appear to be somewhat self-fulfilling. (...)
>
> In this figure we assume that:
>   1) The entropy is a quantity worth maximizing to get good exploration as it can be seen as a measure of how many goals we can reach.
>   2) Our approximation is a good representation - at least we don’t know any way to do better, what did you have in mind with ‘some other analysis of entropy’?
>
> ChronoGEM shows superiority in this experiment but also in the goal conditioned setup - the real end goal - which is the main argument ChronoGEM is superior to the compared baselines.
>
> << Why is this method not compared to a PT and APS? (...)
>
> ChronoGEM is an exploration algorithm, so it could have been compared to all the exploration algorithms we discussed in our related work section (which already mentions APT, APS and UPSIDE).
> Our work is not about improving the SOTA of pure exploration, but rather about building a method to learn a policy that can reach any state. That’s why we decided to implement what we estimated the most commonly used baselines, SMM and RND, rather than the most sophisticated exploration methods.
> Note that SMM +goal-based SAC and RND + goal-based SAC are also our contributions, since the novelty is about the methodology to train goal-reaching agents.
> We would like to note that this work is to our knowledge the first successful application of unsupervised continuous control of humanoid in high dimension (copying the position and the pose).
>
> << The paper proposes a helpful algorithm to allow for training a goal-based agent to reach many goals. The work requires further details and comparisons to prior methods to understand if it is an improvement over prior work.
>
> This work is about training agents to accurately reach any goal. In that regard, the related work that could be compared to C3PO is rather the goal-based RL literature than pure exploration. To the best of our knowledge, we have the best reaching accuracy for continuous control tasks (with agents able to jump in order to match a position in the air), but this is only “visual” as there are no good metrics to compare goal achievement in high-dimensional spaces.

---

> > ### Comment · Reviewer_nd1d · 2022-11-24
> > **Follow up comments**
> >
> > << Why are the XYZ positions of everybody part added to the state
> >
> > This will also make the learning task unnecessarily difficult. Many prior methods that use goal-conditioned policies work much better when some redundant information is hidden. This choice seems designed to make the task more challenging, and the motivation for this increased challenge is not clear and not motivated by downstream uses for a goal-conditioned policy.
> >
> > << The results in figure 4 that compare different methods depending on the measure proposed for entropy over a prior distribution appear to be somewhat self-fulfilling. (...)
> >
> > Entropy may not be the best measure. Training goal-conditioned policy is valuable, however, they are used for downstream tasks. Why does training an entropy-focused objective ensure the downstream usefulness of the policy?
> >
> > << Our work is not about improving the SOTA of pure exploration but rather about building a method to learn a policy that can reach any state.
> >
> > This statement points to some of the concerns about the work. Can out outline the value of training a policy that can reach any state? Typically there are few states worth learning to visit.

---

### Author Response · Authors · 2022-11-18
**General comment.**

We would like to note that this work is to our knowledge the first successful application of unsupervised continuous control of humanoid in high dimension (copying the position and the pose).

This is also, as far as we know, the first goal-reaching policy able to control agent to jump and achieve poses in the air.

---

> ### Comment · Reviewer_nd1d · 2022-11-24
> **Related work**
>
> Two related works on training humanoid models
>
> Peng XB, Berseth G, Yin K, Van De Panne M. Deeploco: Dynamic locomotion skills using hierarchical deep reinforcement learning. ACM Transactions on Graphics (TOG). 2017 Jul 20;36(4):1-3.
>
> is one of the first works to train a humanoid agent with a goal-conditioned policy. Also,
>
> Merel J, Ahuja A, Pham V, Tunyasuvunakool S, Liu S, Tirumala D, Heess N, Wayne G. Hierarchical visuomotor control of humanoids. arXiv preprint arXiv:1811.09656. 2018 Nov 23.
>
> It is not clear how your humanoid control method is the first successful example. Please clarify.

---

### Decision · Program_Chairs · 2023-01-20

**Decision:**

Reject

**Justification For Why Not Higher Score:**

The weaknesses (mentioned above) outweigh the positive aspects of the paper, so the paper cannot be accepted at this point.

**Justification For Why Not Lower Score:**

N/A


**Metareview: Summary, Strengths And Weaknesses:**

The paper proposes a method called C3PO that has two stages. In the first stage, the method explores the state space to reach different goals in the space. In the second stage, the collected data (states) are used as goals, and a goal-conditioned policy is trained to reach those states (goals). Ideally, the proposed method should explore all states uniformly during the exploration phase.

Strengths:

- The problem of exploration is important, and the unsupervised nature of the proposed method makes it more interesting.

- The exploration method is more effective compared to random walks, SMM, and RND for this particular set of tasks.

- The provided examples are nice and illustrative of what the method can achieve.

Weaknesses:

- There are concerns regarding the "infinite sample" budget that the method requires to achieve a uniform distribution.

- The proposed approach is quite sample inefficient, and its applicability is limited to "cheap" simulators.

- There is no comparison with prior SOTA approaches on exploration.

All three reviewers have carefully read the paper and the rebuttal. They did not find the rebuttal convincing and recommended rejection. The AC also read the reviews and the rebuttal and agrees with the reviewers' recommendation.